



# Liquid-water content and water distribution of wet snow using electrical monitoring.

Pirmin Philipp Ebner[1], Aaron Coulin[1], Joël Borner[1], Fabian Wolfsperger[1], Michael Hohl[1], Martin Schneebeli[1]

[1] WSL Institute for Snow and Avalanche Research SLF, Davos Dorf, 7260, Switzerland

*Correspondence to*: Martin Schneebeli (schneebeli@slf.ch)

**Abstract**:

Snow exists in a wide range of temperatures and around its melting point snow becomes a three-phase material. A better understanding of wet snow and the first starting point of water percolation in the seasonal snowpack is essential for snow pack stability, snow melt run-off and remote sensing. In order to induce and measure precisely the liquid water and the corresponding dielectric properties inside a snow sample, an experimental setup was developed. Using microwave heating at 18 kHz allows the use of dielectric properties of ice to enable heat to be dissipated homogeneously through the entire volume of snow. A desired liquid water content inside the snow sample could then be created and analysed in a micro-computer tomography. Based on the electrical monitoring a promising perspective for retrieving water content and water distribution in the snowpack is given. The heating process and extraction of water content are mainly dependent on the morphological properties of snow, the temperature and the liquid water content. The experimental observation can be divided in three different heating processes affecting the dielectric properties of snow for different densities: (1) dry snow heating process up to 0 °C indicating a temperature and snow structure dependency of the dielectric property of snow; (2) wet snow heating at stagnating temperature of 0°C and the presence of uniformed distributed liquid water changes the dielectric properties. The presence of liquid water decreases the impedance of the snow sample until water starts to percolate; and (3) the start of water percolation is between 5-12 water volume fraction depending on the



snow density and confirms the literature findings. The onset of water percolation
initiated an inhomogeneity in snow and water distribution, strongly affecting the
dielectric properties of the snow. These findings are pertinent to the interpretation of
the snow melt run-off of spring snow. These laboratory measurements allow to find
the narrow range of the starting point of water percolation in coarse-grained snow and
to extract the corresponding dielectric properties which is important for remote
sensing.



## 1. Introduction

Snow, a sintered porous material made of ice grains, has a complex porous microstructure and consist of a continuous ice structure at temperature below zero degrees (Löwe et al., 2011). Reaching the melting point, snow becomes a three-component mixture of ice, water and air. A detailed understanding of the influence of liquid water on the snow microstructure is essential. It influences radar and microwave attenuation, sub-surface exploration, remote sensing, radar altimetry, electrical grounding, atmospheric electrical fields and electrostatic charging by precipitation and blown particles (Mellor, 1977). Liquid water in snow is also a critical factor for estimating the hazard of wet snow avalanches and the transmission of melt-water through a snow-pack (Evans, 1965; Reiweger et al., 2015).

Estimating liquid water content is difficult even for experienced observers (Martinec, 1991; Fierz and Föhn, 1994), partly because water flow through snow varies both spatially and temporally (e.g. Colbeck 1979; Marsh, 1988; Conway and Benedict, 1994). Introduction of liquid water into snow changes morphological (Brun, 1989; Coléou and Lesaffre, 1998; Raymond and Tusima, 1979; Brun, 1989; Marsh, 1987; Colbeck, 1997; Marshall et al., 1999; Jordan et al., 2008) and mechanical properties of snow (Techel et al., 2011; Colbeck, 1982). When liquid water occurs for the first time in the seasonal snowpack, much water can be hold as small grains cause high capillary forces. After a couple of melt-freeze cycles smaller grains disappear and less melt-water is held immobile in the snow matrix (Yamaguchi et al., 2010). This delays water runoff during the early stages of snowpack melt-out and snow can retain liquid water (Colbeck, 1972; Linsley et al, 1949). Reaching a saturation point, the liquid water is then released suddenly. The transition point where the liquid water starts to percolate in the snow sample is important because it dominates the spring runoff period in many regions.

Measuring the liquid water content non-destructively and homogeneously in larger snow samples (in centimetre regime) is very challenging. The small amount of liquid water present and the sensitivity of the snow to the various processes to induce a defined liquid water content makes the measurements hard. Radiative absorption or snow melting in a room at 0°C induces a temperature gradient from the surface to the core of the snow sample leading to an inhomogeneous liquid-water content

distribution. The direct supply of liquid water causes percolation in preferential
selected channels. In both cases, the water distribution is not homogeneous and the
exact extraction of the starting point of water percolation is not possible. In contrast,
Coléou and Lesaffre (1998) performed experiments by slowly saturate a snow sample
fully with water and afterwards drained out to find the starting point of water
percolation. They approached the water retention curve of snow (Yamaguchi et al.,
2010) from the right site. In these experiments water percolation was initiated at
around 5-14 % of mass volume for snow density between 350 kg m$^{-3}$ and 680 kg m$^{-3}$
(Coléou and Lesaffre, 1998).
Another way to induce a homogeneous liquid-water content into the snow without
destroying the snow sample is microwave heating (Brun, 1989; Camp and LaBrecque,
1992). In this case, the water retention curve of snow (Yamaguchi et al., 2010) is
approached from the left site. A uniform electric field oscillating at an appropriate
frequency excites the dielectric properties of ice enabling heat to be dissipated through
the whole volume of snow (Mellor, 1977; Brun, 1989; Camp and LaBrecque, 1992).
The dielectric properties of the ice depend on the frequency, temperature and snow
density. The applied field results in a displacement of charged particles in the
insulating material, giving rise to induced dipoles. The permanent dipoles of the water
molecule respond to the electric field, which results in a temperature increase of the
material. The heat absorption by the ice phase is uniform and the absorbed energy
depends only on the imaginary part $\varepsilon$" of permittivity (Polder and Van Santen, 1946).
Dielectric properties of dry snow are closely related to solid ice. The ideal relaxation-
frequency of ice is at 7.5 kHz (Auty and Cole, 1952). However, the relaxation
frequency of dry snow lies between 10 and 100 kHz depending on the snow density
and temperature (Bader and Kuroiwa, 1962; Polder and Van Santen, 1946; Evans,
1965). The presence of liquid water strongly affects the dielectric properties of the wet
snow sample (Sweeny and Colbeck, 1974; Ambach and Denoth, 1980, Camp and
LaBrecque, 1992). A wide spectrum of frequencies has been explored to determine
the free water content of water in snow (Ambach and Denoth, 1975; Boyne and Fisk,
1987; Brun, 1989; Denoth et al., 1984; Denoth and Foglar, 1986; Perla, 1990; Camp
and LaBrecque, 1992). As long as the liquid water phase remains discontinuous on
the ice matrix (Brun E., 1989), the dielectric properties of the wet snow sample are



homogenous over the whole sample and the amount of liquid water can be estimated (Ambach and Denoth, 1972, Koch et al., 2014). At the point where the liquid water starts to percolate, an inhomogeneous distribution of water and ice starts to build up in the total snow sample and locally affects the dielectric properties of the sample. Work by Camp and LaBrecque (1992) showed that dielectric heating at 20 kHz is a useful means of modifying the water content from 0 to 30% by weight.

The objective of this paper is to present an experimental setup to allow standardized studies to extract the starting point of water percolation depending on snow density and dielectric properties. We developed a dielectric heating device at 18 kHz, similar to the work by Brun (1989) and Camp and LaBrecque (1992), to find the narrow range of the starting point of water percolation in coarse-grained snow. We improved the suggested power measurements by improving the measurement of phase to be more sensitive in the control of water content. Our technique of monitoring the voltage, current and phase shift at the two copper plates makes it possible to study the dielectric properties of the snow as the water content changes. In particular, at temperature close to the melting point the surface properties of the ice change markedly affecting the dielectric properties. Additionally, with the well-controlled electrical heating the exact water content of water percolation of different kind of snow densities and surface-to-volume ratio can be extracted. We analysed three regimes: (1) dry snow heating showing the fraction of energy absorption of the snow, (2) producing wet snow to investigate the starting point of water percolation based on the electrical properties and density of the snow. Additionally, primarily quantifying of the water content in three-dimensional space without destroying the snow structure are analyzed using micro computed tomography (micro-CT); (3) water percolation affecting the overall impedance of the snow sample.

**2. Experimental setup**

The experimental setup is shown schematically in Fig. 1 and a photo of the experimental setup is shown in Fig. 2. The device consists of three functional blocks: (1) low voltage circuit to generate the sinusoidal signal and amplify the energy output, (2) high voltage circuit to transform the low primary voltage to a high secondary voltage, and (3) design of the sample holder between the high voltage capacitor plates.





The core of the snow heater is a Red Pitaya STEM 125-14 using for signal generation
and data acquisition, and is controlled via Standard Commands Programmable
Instruments SCPI in Matlab. The low voltage sinusoidal input signal with a frequency
of 18 kHz is generated by a high-speed digital to analog converter and is amplified
afterwards to stabilizes the electrical potential in the circuit. A step-up transformer
transforms the low primary voltage to a high secondary voltage of around 350 V
applied to two copper plates inducing the dielectric heat into the snow sample. The
surface of the copper plates is electrically insulated to prevent Joule heating of the
snow sample. The snow sample was placed into a polyoxymethylene (POM) ring
(diameter = 60 mm, distance = 13 mm) and inserted between the two capacitor plates.
The snow sample and the capacitor are thermally insulated with extruded polystyrene
foam (XPS) with a thickness of 120 mm to prevent radial conductive and convective
heat losses.
The applied sinusoidal waveforms of voltage $U$(t) to the copper-plates is attached to a
differential probe and is measured galvanic sorted with a 100-fold attenuation. The
current $I$(t) from the plate is measured via a shunt resistor. The phase shift $\varphi(t)$
between the sinusoidal waveforms of voltage and current is measured between the
circuit's input and circuit's output signal. An input protection circuit prevent the analog
to digital converter from damage in case of a short circuit. The voltage connection
between the low and high voltage part is measured via a shunt resistor. This
connection defines the star point of the circuit and makes sure that the second part of
the circuit doesn't thrift away. It is the only star point preventing the circuit from circular
currents. A negative temperature coefficient element is placed one centimetre inside
the snow sample to measure the temperature. A low pass filter is applied to block the
noise of the capacitor.
The total power $P_{\mathrm{RMS}}(t)$ between the two copper-plates is calculated based on the root-
mean-squared voltage $U_{\mathrm{RMS}}(t)$, current $I_{\mathrm{RMS}}(t)$ and the measured phase difference $\varphi(t)$
$$P_{\mathrm{RMS}}(t) = U_{\mathrm{RMS}}(t) \cdot I_{\mathrm{RMS}}(t) \cdot cos\varphi(t) \tag{1}$$

The impedance $R_{\mathrm{RMS}}(t)$, describing the resistant of the snow sample, between the two
copper-plates is given by
$$R_{RMS}(t) = \frac{U_{\text{RMS}}(t)}{I_{\text{RMS}}(t)} \qquad (1)$$
The uncertainties of the temperature $T(t)$, current $I_{\text{RMS}}(t)$, voltage $U_{\text{RMS}}(t)$, phase shift
$\varphi(t)$, total power consumed $P(t)$, and density of the snow measured by weighting are:
$\pm 0.05$ °C, $\pm 0.01$ mA, $\pm 0.5$ V, $\pm 2$ degrees, $\pm 0.005$ W and $\pm 20$ kg m$^{-3}$.
**2.1 Tomography experiments**
A cooled micro-computer tomograph (CT; Scanco Medical $\mu$-CT80) at a cold
laboratory temperature of -5 °C was used to visually quantify the water content in
three-dimensional space without destroying the snow structure. The scanned image
had a volume of 200 x 200 x 20 voxels (3.6 mm x 3.6 mm x 0.36 mm) with a nominal
voxel resolution of 18 μm. The grey scale resolution for each voxel was 16 bit and a
Gaussian filter ($\sigma = 1.4$, support = 3) was applied to reconstruct the micro-CT images.
The volume was segmented to a binary image by classifying each voxel by ice or air.
The threshold for the segmentation process was chosen such as that the manually
measured density did not deviate more than 12 % from the CT-density in the
segmentation process (Riche and Schneebeli, 2013). Each scan took around 2.7 h.
Absorption by water and ice are almost identical (Lieb-Lappen et al., 2017), and are
hardly to separate in the segmentation process. Therefore, the water creation on the
snow surface was extracted by superposition of two micro-CT scans. One scan was
taken before the heating process and the second one afterwards. Before the second
micro-CT scan, the wet snow sample was shock frozen at -30 °C to preserve the snow
structure. This allowed us to easily visualize and to extract the water creation on the
surface of the ice matrix with an uncertainty of 4 %.
**3. Method**
The phenomena involved in microwave heating of snow are volumetrically absorption
of electromagnetic energy to achieve self-heating uniformly and rapidly, which is
characterized by the density of the snow. The dielectric power absorption $P$ is equal
to the total power consumed $P_{\text{RMS}}$, given by:
$$P = 2 \cdot \pi \cdot f \cdot E^2 \varepsilon_0 \cdot \varepsilon_s''(f, \rho_s, T_s) \cdot A \cdot d = P_{\text{RMS}} \qquad (3)$$



where $f$ is the frequency, $E = U\,d^{-1}$ the electric field, $\varepsilon_0 = 8.85 \cdot 10^{-12}$ the electric field
constant, $\varepsilon_s''$ the imaginary part of the complex dielectric constant of snow, $A$ the
capacitors surface area and $d$ the distance between the two copper-plates.
Rearranging Eq. (3) the imaginary part of the complex dielectric constant of dry snow
is given by
$$\varepsilon_s''(f, \rho_s, T_s) = \frac{P_{\mathrm{RMS}}}{2 \cdot \pi \cdot f \cdot E^2 \cdot \varepsilon_0 \cdot A \cdot d}$$
(4)

which depends on the frequency $f$, snow density $\rho_s$, and snow temperature $T_s$.
The heating efficiency is an important factor to evaluate the heating process. It is
defined as the ratio of energy absorbed by the heated sample to that radiated from the
microwave source [Ali, 2016] given by:
$$\eta = \frac{Q_{\mathrm{setup}} - Q_{\mathrm{sample}}}{Q_{\mathrm{setup}}} = 1 - \frac{m_s c_p (T_1 - T_0)}{\int_0^{t_1} P_{\mathrm{RMS}}(t)\,dt}$$
(5)

where $m_s$ is the mass of the snow sample, $c_p$ the specific heat capacity, $T_0$ and $T_1$ the
initial temperature and melting temperature at 0 °C, and $t_1$ the time until temperature
reached 0 °C.
The liquid water mass fraction for each timestep $t$, is calculated by the fraction of the
measured dissipated latent heat and total latent heat needed for the phase change:
$$x_{\mathrm{mass}}(t) = \frac{\int_{t_1}^{t} \eta \cdot P_{\mathrm{RMS}}(t)\,dt}{h_{\mathrm{latent}} m_s}$$
(6)

where $h_{\mathrm{latent}} = 334$ kJ kg$^{-1}$ is the latent heat for the phase change from ice to water and
$t_1$ the time step where the snow sample reached 0 °C.
The liquid water volume fraction is given by [Coléou and Lesaffre, 1998]:
$$x_{\mathrm{vol}}(t) = \frac{x_{\mathrm{mass}} \cdot \rho_s}{\rho_i - \rho_s} \cdot \frac{\rho_i / \rho_w}{1 - x_{\mathrm{mass}}}$$
(7)



where $\rho_s$, $\rho_i$ and $\rho_w$ are the snow, ice (917 kg m$^{-3}$) and water (999.9 kg m$^{-3}$) density.
The uncertainties of $x_{mass}$ and $x_{vol}$ are 10 % due to the uncertainty of the power ($\pm$0.005
W) and density measurement ($\pm$20 kg m$^{-3}$).
**4. Results**
Deionized water with a conductivity of $\approx$ 0.2 $\mu$S cm$^{-1}$ was used to produce natural
identical snow (Schleef et al., 2014) in a cold laboratory at -20 °C. The produced snow
was sieved into sample holders (mesh size: 2 x 2 mm) and was sintered at a
temperature of -2 °C for two to five days to allow the snow crystals to form a uniform
grain size. A hydraulic press compressed the snow to densities between 400 and 600
kg m$^{-3}$ to represent snow packs in spring (Bartelt and Lehning, 2002). We analysed in
total seven different snow samples.
The measured electrical properties between the two copper-plates were strongly
influenced by the temperature, water content, and density of the snow sample. The
higher the snow density and the water content in the snow was, the stronger the
measured electrical properties were affected, shown in Table 1. Figure 3 shows a
typical measured temperature $T(t)$, current $I_{RMS}(t)$, voltage $U_{RMS}(t)$, and phase shift $\varphi(t)$
profile of a heating process for snow density of (a) 438 kg m$^{-3}$, (b) 539 kg m$^{-3}$, (c) 612
kg m$^{-3}$, and (d) 917 kg m$^{-3}$. The temperature profile shows the characteristic of the
snow heating process increasing from -1 °C up to 0 °C. Afterwards the temperature
stagnates at 0 °C and the supplied energy was used for the phase change from ice to
liquid water. The current profile has a different behaviour. It shows a slightly linear
increase until the snow sample reached a temperature of 0 °C. Afterwards the incline
of the current curve further increased reaching the highest current of 1.7 mA, 2.6 mA,
3.8 mA, and 4.3 mA at around 80 min, 60 min, 55 min, and 9 min for snow with
densities of 438 kg m$^{-3}$, 539 kg m$^{-3}$, 612 kg m$^{-3}$, and 917 kg m$^{-3}$. After this maximum
the current started to decrease with time. The voltage and phase shift showed a mirror
inverted behaviour to the current profile. Both parameters decreased with time and
increased afterwards again. At the beginning a phase shift of 55.9°, 52.8°, 50°, and
46.9° were measured with the lowest phase shift of 41.3°, 35.2°, 31.5°, and 22.4° for
snow densities of 438 kg m$^{-3}$, 539 kg m$^{-3}$, 612 kg m$^{-3}$, and 917 kg m$^{-3}$.



The snow temperature, density and water content strongly affected the impedance
and the total power consumed by the snow sample. The impedance decreased with
increasing temperature, water content, and density, vice versa for the total power
consumed, shown in Table 2. Figure 4 shows a typical calculated total power $P_{RMS}(t)$
and impedance $R_{RMS}(t)$ profile compared with the measured temperature profile of a
heating process for snow density of (a) 438 kg m$^{-3}$, (b) 539 kg m$^{-3}$, (c) 612 kg m$^{-3}$, and
(d) 917 kg m$^{-3}$. The impedance had the same profile behaviour like the phase shift
starting with 368.4 kΩ, 275.5 kΩ, 213.5 kΩ, and 123.1 kΩ reaching a minimum of
197.4 kΩ, 127.3 kΩ, 87.3 kΩ, and 73.5 kΩ after 80 min, 60 min, 55 min, and 9 min for
snow density of 438 kg m$^{-3}$, 539 kg m$^{-3}$, 612 kg m$^{-3}$, and 917 kg m$^{-3}$. The total power
consumption profile was mirror inverted. It started with 0.17 W, 0.25 W, 0.33 W, and
0.63 W and reached a maximum of 0.41 W, 0.69 W, 1.04 W, and 1.24 W after 80 min,
60 min, 55 min, and 9 min for snow density of 438 kg m$^{-3}$, 539 kg m$^{-3}$, 612 kg m$^{-3}$, and
917 kg m$^{-3}$.
The heating efficiency was affected by heat loss at the wall and decreases with higher
snow density. The microwave power did not directly penetrate into the snow samples
but also through the air space of the pores. As a result, the reflection of microwave
power on the interface, which was caused by the relative permittivity mismatch
between the air and the sample led to limited heating efficiency. As the frequency of
18 kHz was in the range of the optimal snow heating frequency between 10 and 100
kHz depending on the snow density (Bader and Kuroiwa, 1962; Polder and Van
Santen, 1946; Evans, 1965), the efficiency of the heating samples was usually higher
for lower density. This effect is confirmed by Fig. 5 showing the heating efficiency and
the complex dielectric constant of dry snow at $T = 0$ °C for various snow sample. The
error bars indicate the measured uncertainty of the experimental setup. Ice had the
lowest heating efficiency with the highest extracted permittivity value of $\varepsilon_i'' = 30.65$,
similar to literature values of $\varepsilon_i'' = 30.93$ at 18 kHz [Fujita et al., 2000].
The start of water percolation was between 5-12 water volume fraction depending on
the snow density. Dense snow absorbed more microwave energy leading to higher
liquid water content in a snow sample in a short time. The temporal evolution of the
liquid water mass and volume fraction based on the measured power for the different
snow samples (Fig. 6) increased with time and the influence of snow density was




observed. At the maximum of consumed power $P_{RMS}(t)$ a water mass and volume
fraction of 7.1 and 6.1, 6.7 and 8.8, 8.7 and 15.4, and 0.4 and 0 for snow density of
(a) 438 kg m$^{-3}$, (b) 539 kg m$^{-3}$, (c) 612 kg m$^{-3}$, and (d) 917 kg m$^{-3}$ was reached. Table
3 shows the estimated water mass and volume fraction and time of the different snow
sample at the reversal point of the measured power, indicating the start of water
percolation.
The created water led to a stronger rounding of the ice crystals and the effect of fast
wet snow metamorphism was observed where a growth in size of snow crystals was
identified (Colbeck and Davidson, 1973). Figure 7 shows preferential spots of water
accumulation inside the snow structure. Blue indicates the ice structure and orange
shows the water part. The snow sample with an initial density of around 682 kg m$^{-3}$
was heated to induce a volume water content of 16 %. A volume water content of
around 13.3 % was extracted from the difference in the micro-CT scans before and
after the experiments. This rounding effect is also shown in an insight in the snow
heating process by a microscopy image illustrated in Fig. 8. The photography on the
left shows a snow structure of dry snow before and on the right after the experimental
run.
**4. Discussion**
Our major experimental results are summarized in Fig. 4 and 6, and in Table 2 and 3.
The dominant sources of absolute errors in the measurement of the water mass and
volume fraction in the snow were the snow density and the inaccuracy in the power
measurement. Especially, at snow densities below 450 kg m$^{-3}$ this might cause
deviations of $\pm$ 1 % in the water mass and volume fraction measurement. However, at
higher snow densities the relative errors were considerably less.
The snow structure and the water content had a major impact on the electrical
properties showing the same behaviour like in the work of Camp and LeBraque (1992).
In both works the electrical power increased with increasing water content and drops
at one point again. Based on the findings we divided the heating process in three
areas, shown in Fig. 9:



**(1) Dry snow**: The heating process up to 0 °C indicated a temperature and snow structure dependency on the measured values. As snow temperature reached the melting point, the surface properties of the ice structure changed markedly and affected the electrical properties. The vibration and the mobility of protons enhanced and influenced the electrical conductivity leading to a decrease of the impedance. Further, at higher density the structural connections between ice crystals were less destructed by the pore volumes. This allowed a higher rate of flow of electric charge leading to a higher electric current. Additionally, the electrical potential between the two copper-plates was less affected by the pore volume leading to a more stable voltage and smaller phase shift between voltage and current. As a result, the electrical conductivity increased resulting in a lower impedance and a higher electrical energy transfer.

**(2) Wet snow:** Snow was becoming a particularly complicated medium because the introduction of liquid water caused rapid changes of the important material properties. The temperature stagnated at 0°C and the presence of uniformed distributed liquid water changed the dielectric properties of the snow sample. Additionally, the liquid water layer at the surface allowed the mobility of protons resulting in stronger rate of flow of electric charge and therefore enhanced the electrical conductivity. This reduced the impedance of the two-phase material significantly leading to a decrease of the impedance and phase shift, and an increase in electric current and power.

**(3) Water percolation:** The water started to percolate and liquid water accumulated at the bottom of the sample holder. The missing water in the upper part of the sample holder treasured up at the bottom of the sample holder and left empty spots at the top where the density decreased locally. The snow probe was not homogeneous anymore leading to a decrease in the electrical conductivity. As a result, the impedance increased again and the electric power decreased. First camera picture of water percolation after an experimental run is shown in Fig. 10. The sample holder was aligned vertically between the capacitor plates. Water percolated in the upper part of the sample and accumulated at the bottom of the sample holder leading to an inhomogeneous mixture of the sample. This inhomogeneous mixture changed the dielectric properties of the complete sample and affected the heating process. After



this state the relative error of the water mass and volume fraction calculation
increased.
Based on the findings, water percolation occurred over a narrow range of values in
coarse-grained snow (see Table 3) and was initiated at around 5-8 % of the mass
volume (see Table 3). For high snow densities where the surface-to-volume ratios
were small, our results were lower than found by Coléou and Lesaffre (1998).
Following reasons are: (1) they approached the retention curve of snow (Yamaguchi
et al., 2010) from the opposite site and therefore the physical processes were different,
(2) they fully saturated the snow sample for about 5 minutes. Therefore, the surface
tension of water had an additional effect, like a suction effect, holding more water in
the pore space. In our approach water could not be held immobile as the percolation
started earlier at the smooth ice surface.
Although the micro-CT measurements (see Fig. 7) showed a snow sample after the
water percolation point, still preferential spots of water accumulation inside the snow
structure could be seen. Three interesting observations were visible after percolation:
(1) No water film around the snow structure but isolated smaller and larger water
accumulations were visible indicating that phase change from ice to water were
happening on preferential spots on the ice crystal. However, it has to mention that the
pixel resolution was too coarse to detect an additional thin water film around the snow
structure. The created water led to a stronger rounding of the ice crystals and the
dendritic structure further disappeared. Nevertheless, no big change in grain shape
was observable due to the high density. (2) The gravity had no influence on the
orientation of the accumulated water on the ice crystal. It is apparent that the water
was uniformly distributed on the single ice crystals. The water droplets were too small
to be distracted by the gravity. (3) Single water accumulation links between single
neighbouring ice crystals can be seen. The refreezing of the snow sample after the
experiment led to single crystals agglomeration and a growth in size of snow crystals
(Colbeck and Davidson, 1973).
The electrical heating procedure developed to incrementally melt snow in order to vary
the water content and to analyse the created water non-destructive in a micro-CT
worked very well. Improving the experimental setup that the frequency can be
increased to the GHz-MHz regime for a short period of time, the exact dielectric snow



property based on the snow morphology and water content can be extracted. This will
allow to improve remote sensing and field measurements on the snow-water-
equivalent (Ambach and Denoth, 1972, Koch et al., 2014).
**5. Summary and Conclusion**
We designed, fabricated, and tested an experimental setup for in-situ time-lapse
nondestructive investigation of water percolation in snow using the electrical
properties of snow. Frequency heating close to the relaxation frequency of ice was
applied to slowly increase the water content uniformly in the snow sample until liquid
water started to percolate. By measuring the temperature and the applied power, the
water content in the snow sample at each timestep was deduced. This new instrument
allows to elucidating the starting point of water percolation based on measured
electrical and morphological properties of the snow. The setup and the obtained
results can be used to precisely forecast the run-off time of different density
snowpacks and to investigate the mechanical properties, water movements, surface
friction, adhesion, and liquid-water measurements, for wet snow and ice.
The experimental observation showed three different heating processes affecting the
dielectric properties of snow for different densities: (1) dry snow heating process up to
0 °C indicating a temperature and snow structure dependency of the dielectric property
of snow. At warmer temperature, slightly higher complex dielectric constant were
measured having higher discrepancy for more dense snow; (2) wet snow heating at
stagnating temperature of 0°C and the presence of uniformed distributed liquid water
changes the dielectric properties and therefore reduces the impedance of the two-
phase material significantly until the starting point of water percolation; and (3) the
start of water percolation is between 5-12 water volume fraction depending on the
snow density. After this point the snow sample has an inhomogeneous mixture where
liquid water treasures up at the bottom of the sample holder and is leaving bigger
pores in the upper part leading to an increase of overall impedance of the snow
sample.
Our results and conclusions indicate that there is a need for additional validation.
Specially, it would be crucial to not only look at the density but also at the specific
surface area of the snow at a given density which also affects the capillary forces and



therefore the starting point of water percolation. Ideally, the entire snow sample will be
tomographically measured before the experiment to extract the morphological
parameters. The primarily micro-computer tomography (CT) result (Fig. 7) shows first
promising visualization of the preferred spots of liquid water in three-dimensional
space without destroying the snow structure. However, more detailed measurements
are needed to make stronger statements about preferential spots of water
accumulations inside the snow sample.

**Acknowledgments:**
The authors thank H. Loewe and B. Walter for the constructive reviews and the
modelling support.



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



**Table 1:** Density of the snow samples and the corresponding voltage $U_{RMS}$, current $I_{RMS}$, and phase shift $\varphi_{RMS}$ at the start of the experiment (init), reaching 0 °C (dry-wet) and the point where the power is maximum (peak).

| Density (kg m$^{-3}$) | $U_{RMS}$ (V) | | | $I_{RMS}$ (mA) | | | $\varphi_{RMS}$ (°) | | |
|---|---|---|---|---|---|---|---|---|---|
| | init | dry-wet | peak | init | dry-wet | peak | init | dry-wet | peak |
| 427 | 325.5 | 325.6 | 326.9 | 0.81 | 0.82 | 1.21 | 57.4 | 57.3 | 48.7 |
| 438 | 332.6 | 327.2 | 328.0 | 0.90 | 0.90 | 1.66 | 55.9 | 55.7 | 41.4 |
| 465 | 328.0 | 328.3 | 328.1 | 1.14 | 1.16 | 1.83 | 52.2 | 52.2 | 40.6 |
| 465 | 329.4 | 330.1 | 329.9 | 1.10 | 1.13 | 1.81 | 52.4 | 52.1 | 40.7 |
| 539 | 339.1 | 329.3 | 327.8 | 1.23 | 1.29 | 2.58 | 52.8 | 51.0 | 35.2 |
| 612 | 331.3 | 332.0 | 327.0 | 1.55 | 1.65 | 3.76 | 49.5 | 48.8 | 31.5 |
| 917 | 336.5 | 327.9 | 313.8 | 2.73 | 3.41 | 4.27 | 46.9 | 34.5 | 22.4 |




**Table 2:** Density of the snow samples and the corresponding impedance $U_{RMS}$ and power $P_{RMS}$ at the start of the experiment (init), reaching 0 °C (dry-wet) and the point where the power is maximum (peak).

| Density (kg m⁻³) | $R_{RMS}$ (kΩ) | | | $P_{RMS}$ (W) | | |
|---|---|---|---|---|---|---|
| | init | dry-wet | peak | init | dry-wet | peak |
| 427 | 403.4 | 396.9 | 269.4 | 0.14 | 0.14 | 0.26 |
| 438 | 368.4 | 362.0 | 197.5 | 0.17 | 0.17 | 0.41 |
| 465 | 287.8 | 282.4 | 179.2 | 0.23 | 0.23 | 0.46 |
| 465 | 300.4 | 291.1 | 182.3 | 0.22 | 0.23 | 0.45 |
| 539 | 275.5 | 254.8 | 127.3 | 0.25 | 0.27 | 0.69 |
| 612 | 213.4 | 213.4 | 87.3 | 0.33 | 0.36 | 1.04 |
| 917 | 123.1 | 123.1 | 73.5 | 0.63 | 0.92 | 1.24 |




**Table 3:** Density of the snow samples and the corresponding heating time and the
water mass and volume fraction where water starts to percolate.

| Density (kg m$^{-3}$) | Heating time (min) | Water mass fraction (%) | Water volume fraction (%) |
|---|---|---|---|
| 427 | 94.5 | 4.1 | 3.3 |
| 438 | 81.1 | 6.4 | 5.2 |
| 465 | 51.2 | 4.3 | 4.1 |
| 465 | 55.2 | 4.6 | 4.2 |
| 539 | 58.2 | 5.8 | 7.3 |
| 612 | 54.5 | 7.5 | 12.9 |
| 917 | 8.79 | 0.3 | 0 |





**Figure captions:**

**Figure 1:** The experimental setup consisting of three functional blocks (1) low voltage circuit to generate the sinusoidal signal and amplify the energy output, (2) high voltage circuit to transform the low primary voltage to a high secondary voltage, and (3) design of the sample holder between the high voltage capacitor plates.

**Figure 2**: (Top) Illustration of the snow heating device. The setup includes a function generator, an audio amplifier and a plastic box with all the high voltage parts. The lid of the box is secured by a safety switch. (Bottom) An illustration of the inner part of the box is shown. It illustrates the high voltage parts with the 60 mm capacitor. Additionally, the CT sample holder with the 34 mm capacitor is shown.

**Figure 3:** Typical measured temperature $T$(t), current $I_{RMS}$(t), voltage $U_{RMS}$(t), and phase shift $\varphi$(t) profile of a heating process for snow density of (a) 438 kg m$^{-3}$, (b) 539 kg m$^{-3}$, (c) 612 kg m$^{-3}$, and (d) 917 kg m$^{-3}$.

**Figure 4:** Typical measured temperature $T$(t), impedance $R_{RMS}$(t) and power $P_{RMS}$(t), profile of a heating process for snow density of (a) 438 kg m$^{-3}$, (b) 539 kg m$^{-3}$, (c) 612 kg m$^{-3}$, and (d) 917 kg m$^{-3}$.

**Figure 5:** Heating efficiency and the complex dielectric constant of dry snow at $T = 0$ °C for various snow sample.

**Figure 6:** The temporal evolution of the liquid water mass and volume fraction based on the measured power for snow density of (a) 438 kg m$^{-3}$, (b) 539 kg m$^{-3}$, (c) 612 kg m$^{-3}$, and (d) 917 kg m$^{-3}$.

**Figure 7:** 3D micro-computer tomography (CT) picture to visualize the water content in snow after water percolation. The scanned image has a volume of 200 x 200 x 20 voxels (3.6 mm x 3.6mm x 0.36mm). Blue indicates the ice structure and orange shows the water part.

**Figure 8:** Photography under the microscope to illustrate the liquid water content in the wet snow sample: (left) snow structure of snow before the experimental run, (right) the same snow sample after 2 hours with an estimated liquid-water content of 12 wt%.



**Figure 9:** Dividing the heating process in three different processes: (1) dry snow heating process up to 0 °C indicating a temperature and snow structure dependency on the measured values; (2) wet snow heating at stagnating temperature of 0°C and the presence of uniformed distributed liquid water changes the dielectric properties of the snow sample; and (3) starting point of water percolation in the snow sample introducing an inhomogeneous mixture where liquid water treasures up at the bottom of the sample holder and is leaving bigger pores in the upper part.

**Figure 10:** Visualization of water percolation after an experimental run. The sample holder was aligned vertically between the capacitor plates. Water percolated in the upper part of the sample and accumulated at the bottom of the sample holder leading to an inhomogeneous mixture of the sample affecting the heating process of the sample.




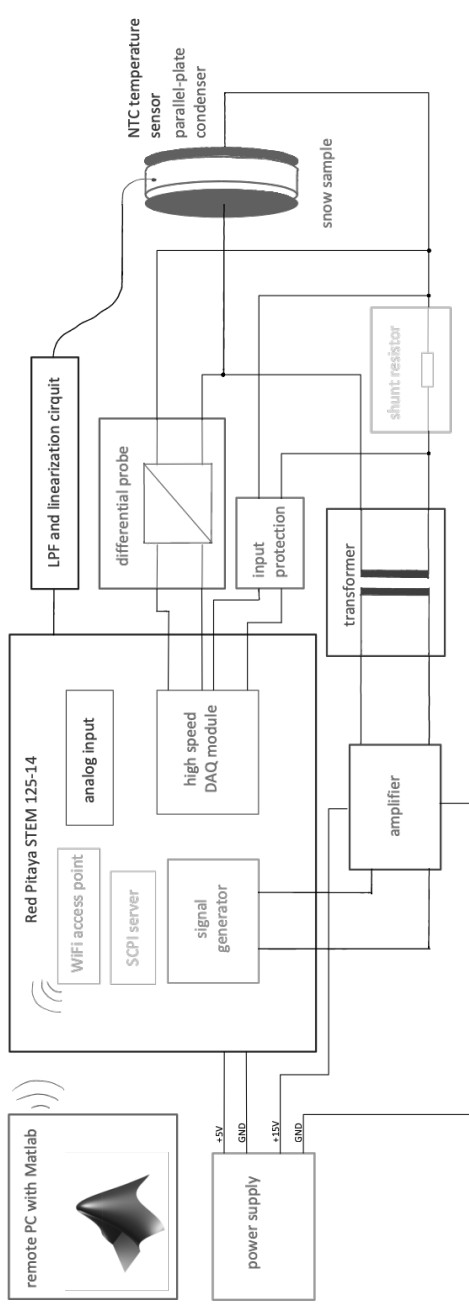


595                                    Figure 1



Figure 2






a)

b)

c)

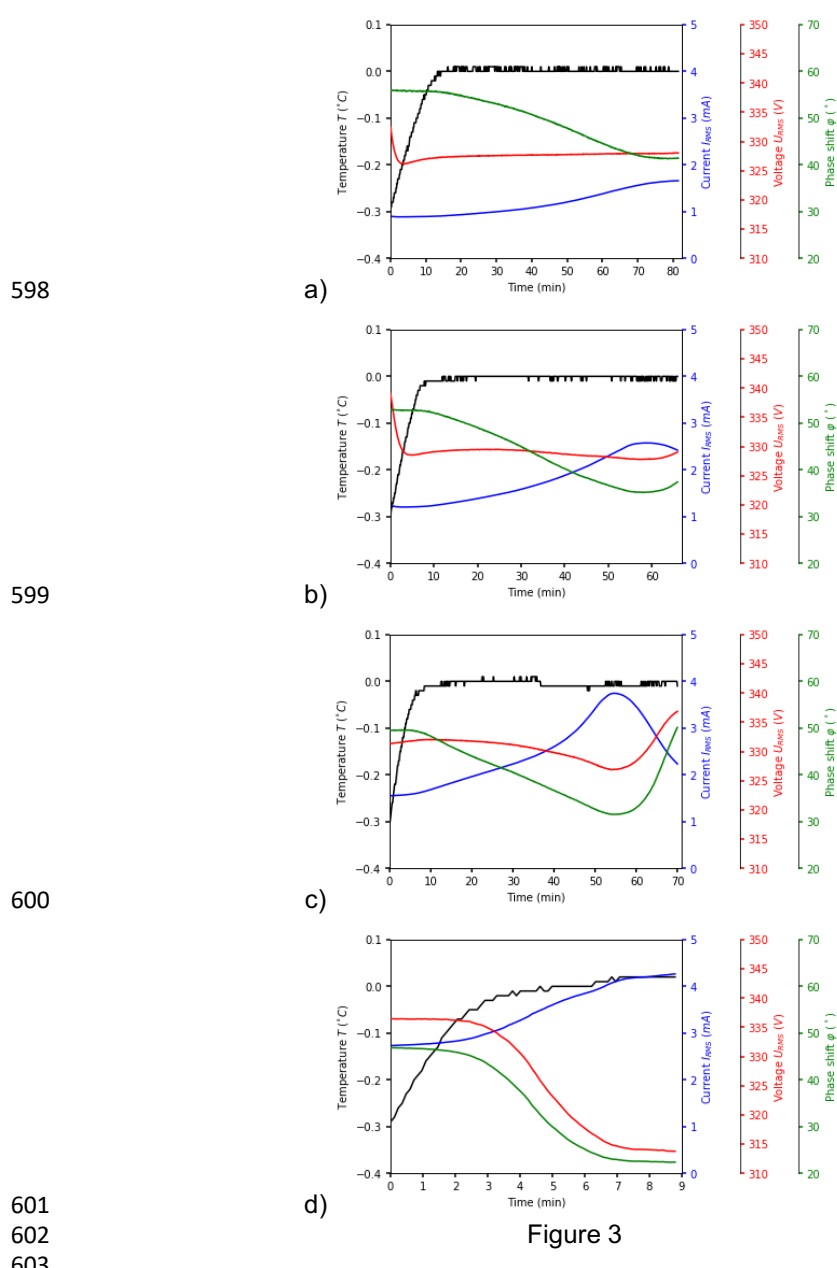

601       d)
Figure 3





a)

b)

c)

607          d)
Figure 4



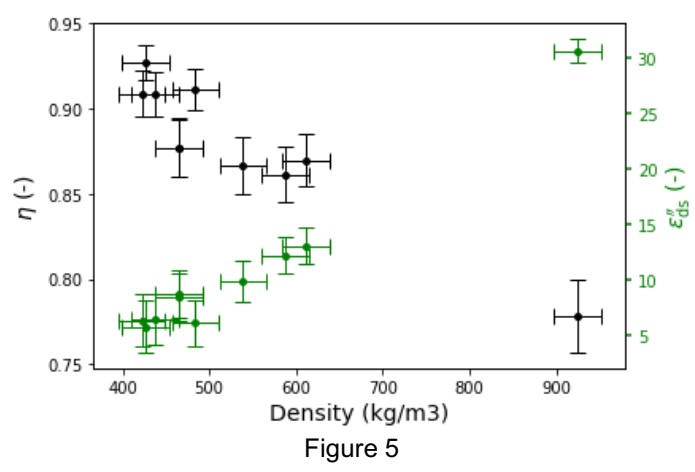


Figure 5




a)

b)

c)

616          d)

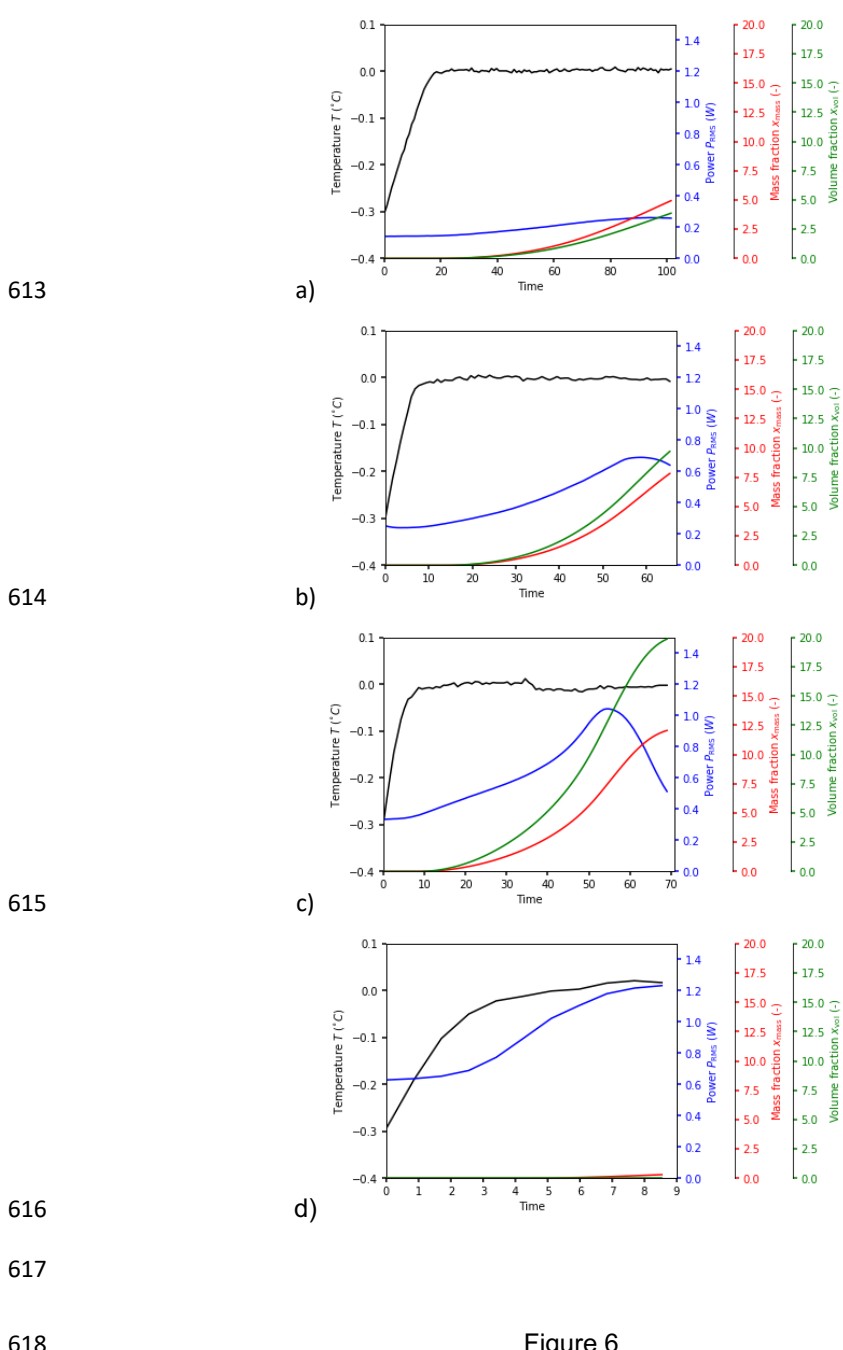


618                          Figure 6



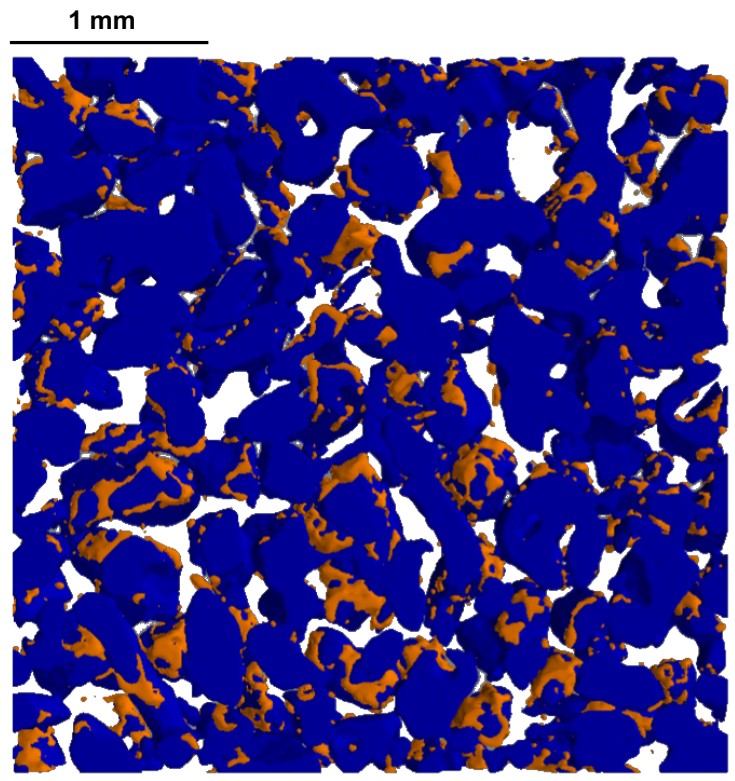

621          Figure 7




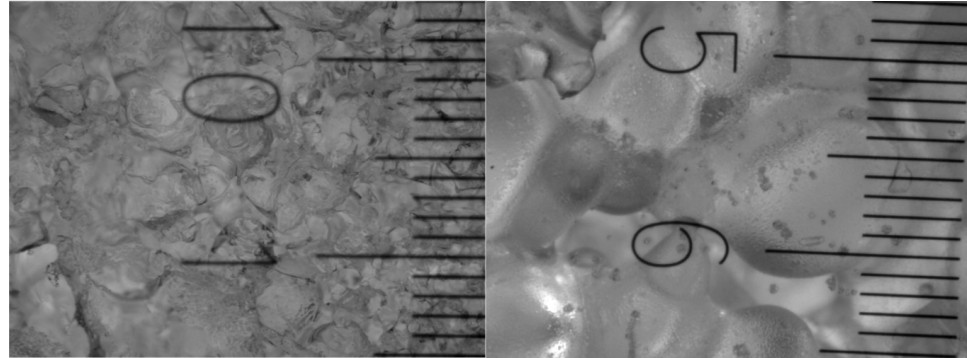

Figure 8





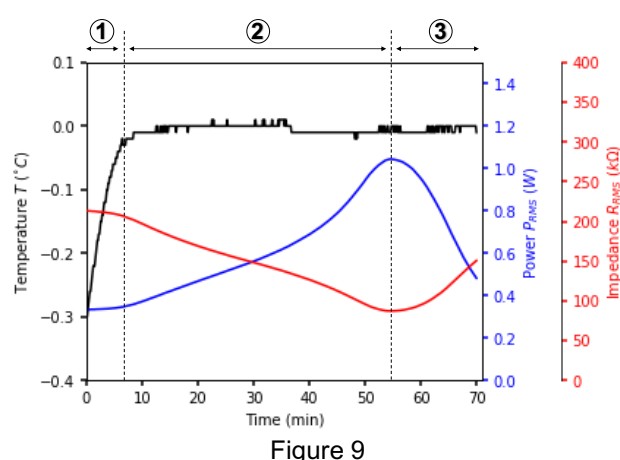


Figure 9



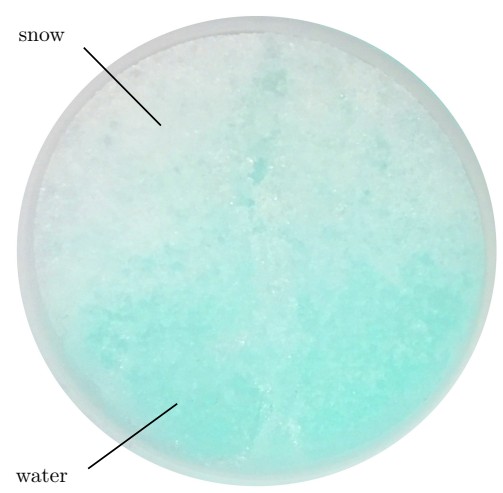

633         Figure 10