# Peer review of "Liquid-water content and water distribution of wet snow using electrical monitoring."

_The Cryosphere, 2020_

## Referee Comment (RC1) · Anonymous Referee #1 · 2 Jun 2020

The authors developed an experimental setup to investigate the liquid water and dielectric properties of snow and reported on the cold room experiment results using their experimental setup. In this manuscript, they classified the heating process of the snow into three phases: Dry snow, Wet snow, and Water percolation based on the characteristics of electric properties. They also showed the potential of micro-CT analyses for wet snow science. I think the manuscript is designed well and its scientific content is enough for publication in The Cryosphere. Before acceptance, several points, which are shown in comments, should be considered.

<Comments>

Abstract:

L33: Please add several concrete explanations how to help your result for the interpretation of the snow melt run-off of spring snow.

1. Introduction:

L76 and L 82: The authors used the sentences "water retention curve of snow from right(left) side", but it is hard to image what is right(left) side. Please add several detailed explanations for right(left) side.

L120: "surface-to-volume ratio" means "Specific surface area"?

2. Experimental setup

Although the sentence of micro-CT experiment was named "2.1 Topography experiments", the first sentence to show the measurement experiment of dielectric properties of snow during heating is not named. I recommend to name the fast sentence likely the sentence of micro-CT part.

L165: How to estimate the uncertainties of measurements? Please add more detail explanations.

3. Method

P8 L198 and L201: $\eta$ should be heating efficiency, please clearly indicate what is $\eta$ in the text.

4. Results

L 224-L226: The authors say that Table 1 shows that "The higher the snow density and the water content in snow was, the stronger the measured electrical properties were affected. But I can not agree to their argument because Table 1 does not show any information of water contents. To clear the evidence of their argument, please add the information of water content in Table 1 or add several explanations in the text how to get the information of water content from the information of the current version of Table 1.

L229-L230: Although text specifies the range of temperature from -1 to 0 °C, the temperature range in Fig. 3 is from -0.4 to 0 °C. The range description should be unified between text and figure.

L237-L237: Although the authors insist on that "After this maximum the current started to decrease with time", I can not agree to their argument because the current graphs of 438 kg m-3 and 917 kg m-3 do not show such trends, namely they only seem to increase during the period in Fig. 3.

L238-L239: Although the authors insist on that "Both parameters decrease with time and increased afterwards again", I can not agree to their argument because voltage and phase shift of 917 kg m-3 does not show such trends, namely they only seem to decrease during the period in Fig. 3.

L248-L252. Although the authors insist on that "The impedances of 38 kg m-3 and 917 kg m-3 reached minimum values after 80 min and 9 min respectively, I can not agree to their argument because the impedance graphs of 438 kg m-3 and 917 kg m-3 still seem to continue decreasing during the period in Fig. 4.

4 Discussion

The number of Discussion part should be 5.

L295-L297: Please add the description how to calculate the deviations.

L308-L309: It is difficult to understand the sentence that "at higher density the structural connections between ice crystals were less destructed by the pore volume". Please add more detailed description.

L337: The range of mass water volume when water percolation started in Table 3 is from 4.1 to 7.5 %, therefore, the description in the text (5-8%) had better be 4 -8 %.

L545: URMS should be RRMS.

---

## Author Comment (AC1) · 15 Jun 2020

**RESPONSE TO ANONYMOUS REVIEWER #1 TO MANUSCRIPT tc-2020-56-RC1**

- *Title:* Liquid-water content and dielectric properties of wet snow using microwave heating
- Authors: Pirmin Philipp Ebner, Aaron Coulin, Joël Borner, Fabian Wolfsperger, Michael Hohl, and Martin Schneebeli

We thank the anonymous referee #1 for his constructive comments and suggestions. All line numbers correspond to the discussion paper and all added texts to the discussion paper are marked red.

**ANONYMOUS REVIEWER #1**

The authors developed an experimental setup to investigate the liquid water and dielec- tric properties of snow and reported on the cold room experiment results using their experimental setup. In this manuscript, they classified the heating process of the snow into three phases: Dry snow, Wet snow, and Water percolation based on the charac- teristics of electric properties. They also showed the potential of micro-CT analyses for wet snow science. I think the manuscript is designed well and its scientific content is enough for publication in The Cryosphere. Before acceptance, several points, which are shown in comments, should be considered.

Comment #1: L33: Please add several concrete explanations how to help your result for the interpretation of the snow melt run-off of spring snow.

[ANSWER] We changed the sentence to:

L33: "These findings are pertinent to the interpretation of the snow melt run-off of spring snow as the restrained amount of water in the snowpack can be extracted and the expected amount of water run-off into the rivers can be calculated."

**Comment #2:** L76 and L 82: The authors used the sentences "water retention curve of snow from right(left) side", but it is hard to image what is right(left) side. Please add several detailed explanations for right(left) side.

[ANSWER] We changed both sentences to:

L76: "... water retention curve of snow (Yamaguchi et al., 2010) from the wetting curve"

L82: "... water retention curve of snow (Yamaguchi et al., 2010) is approached from the dry curve."

Comment #3: L120: "surface-to-volume ratio" means "Specific surface area"? [ANSWER] Yes, we changed it accordingly.

Although the sentence of micro-CT experiment was named "2.1 Topography experiments", the first sentence to show the measurement experiment of dielectric properties of snow during heating is not named. I recommend to name the fast sentence likely the sentence of micro-CT part.

[ANSWER] We changed the sentence in L128 to:

L128: "... and a photo of the high voltage part of experimental setup and the micro-CT sample holder is shown in Fig. 2." And change Fig. 2 to:

**Figure 2**: An illustration of the inner part of the box is shown. It illustrates the high voltage parts with the 60 mm capacitor. Additionally, the micro-CT sample holder with the 34 mm capacitor is shown.

**Comment #4:** L165: How to estimate the uncertainties of measurements? Please add more detail explanations.

[ANSWER] The uncertainties of measurement are given by the measurement equipment and its measurement channel configuration. The current and voltage are measured with the high-speed channel of the Red Pitaya STEM 125-14, but with different channel configurations. The temperature is measured with an NTC attached to a linearization circuit (page 1076 of Tietze, Schenk, Gamm: Halbleiter-Schaltungstechnik 14.Auflage, ISBN:978-3-642-31025-6) with integrated low pass filter and then measured via low speed adc channel on Red Pitaya STEM 125-14. The other parameters are the calculated form these measurement values.

Detailed information:

Measurement error voltage:

Diff probe 100-fold attenuation (manufacturer and type unknown) DC Offset at HV is  $\pm 5 \text{ mV}$

(https://redpitaya.readthedocs.io/en/latest/developerGuide/125-14/fastIO.html)

With 100-fold attenuation the result is: ±500 mV

Measurement error current:

 $\pm 0.5$  mV at LV; measured resistor is  $10\Omega$

 $\Delta I = \Delta V \cdot R = \pm 0.5 \text{ mV} \cdot 10 \Omega = \pm 0.05 \text{ mA}$

Measurement error power:

 $\Delta P = \Delta U \cdot \Delta I = \pm 500 \text{ mV} \cdot 0.05 \text{ mA} = \pm 25 \text{ mW}$

Phase accuracy:

Depending on measuring frequency and signal frequency Measuring frequency: 125000000 Hz / 64=1'953'125 Hz Signal frequency: 18000 Hz

Samples pro  $360^{\circ} = f_{sample} / f_{signal} = 1953125 \text{ Hz} / 18000 \text{ Hz} = 108.507$  $\Delta \varphi = \pm 0.5 \text{ Samples} = 0.5 \cdot 360^{\circ} / 108.507 = 1.659^{\circ}$

Measurement error temperature:

The uncertainty of  $\pm 0.05 \, \, ^\circ \! C$  of the temperature sensor was measured in ice water.

Measurement error density:

The uncertainty of the scales was around 0.7 gr.

 $\Delta \rho = 0.7 \text{ gr} / (\pi \cdot r^2 \cdot h) = 0.7 \text{ gr} / (\pi \cdot (30 \text{ mm})^2 \cdot 13 \text{ mm}) \approx \pm 20 \text{ kg m}^{-3}$ We added following sentences to L165:

L165: "The uncertainties of the temperature T(t) (measured in ice water), current  $I_{RMS}(t)$ , voltage  $U_{RMS}(t)$ , phase shift  $\varphi(t)$ , total power consumed  $P_{RMS}(t)$ , and mass of the snow sample measured by weighting are:  $\pm 0.01 \, ^\circ\text{C}$ ,  $\pm 0.05 \, \text{mA}$ ,  $\pm 0.5 \, \text{V}$ ,  $\pm 1.66$  degrees,  $\pm 0.025 \, \text{W}$  and  $\pm 0.001 \, \text{kg}$ . The uncertainties of measurement are given by the measurement equipment and its measurement channel configuration. The current and voltage are measured with the high-speed channel of the Red Pitaya STEM 125-14, but with different channel configurations. The temperature is measured with an NTC attached to a linearization circuit with integrated low pass filter and then measured via low speed adc channel on Red Pitaya STEM 125-14. Further uncertainty calculations are based on these uncertainties (see Appendix). "

**Comment #5:** L198 and L201:  $\eta$  should be heating efficiency, please clearly indicate what is  $\eta$  in the text.

[ANSWER] We changed the sentence:

L198: "The heating efficiency  $\eta$  is an important factor to evaluate the heating process and is the fraction of energy that is absorbed by the sample."

**Comment #6:** L 224-L226: The authors say that Table 1 shows that "The higher the snow density and the water content in snow was, the stronger the measured electrical properties were affected. But I can't agree to their argument because Table 1 does not show any information of water contents. To clear the evidence of their argument, please add the information of water content in Table 1 or add several explanations in the text how to get the information of water content from the information of the current version of Table 1.

[ANSWER] We changed the sentence:

L224: "The measured electrical properties between the two copper-plates were strongly influenced by the temperature and density of the snow sample. The higher the snow density in the snow was ..."

**Comment #7:** L229-L230: Although text specifies the range of temperature from -1 to 0 °C, the temperature range in Fig. 3 is from -0.4 to 0 °C. The range description should be unified between text and figure.

[ANSWER] We changed it accordingly: L229: "... increasing from -0.3 °C up to 0 °C."

Comment #8: L237-L237: Although the authors insist on that "After this maximum the current started to decrease with time", I can't agree to their argument because the current graphs of 438 kg m-3 and 917 kg m-3 do not show such trends, namely they only seem to increase during the period in Fig. 3.

**[ANSWER]** The value 438 kg m-3 was wrong. We had to change it to 427 kg m-3 and changed Figure 3a and 4a. We also added a sentence about the values for ice (917 kg m-3) and added a new picture to Figure 10.

L241: "The ice sample (917 kg m-3) already broke into pieces (Fig. 10 (right)) before reaching the maxima and minima in the current and phase shift measurements."

---

## Referee Comment (RC2) · Maurine Montagnat (Referee) · 28 Jul 2020

This paper provides interesting results about the progressive wetting of snow during heating at 0°C. To do so, dielectric properties are measured during microwave heating, and these properties enable to follow the evolution of heating during 3 stages, dry snow heating, wet snow heating, and the early stage of percolation. Thanks to the experimental configuration, these measurements enable to provide a precise analysis of the percolation initiation stage, and the corresponding water content for various initial snow densities. This seem to be the main result of the paper. An observation of the wet snow sample is done by micro computed X-ray diffraction tomography, but the results are only very briefly analysed.

Since my knowledge regarding wet snow, percolation, and dielectric measurements of water content is very weak, I am not able to estimate the relevance of the results provided, as regard to the concerned community, and to previous work. I hope that the editor or the other reviewer will be able to do so in order to complement my review work.

Regarding the way the results are presented and the organisation of the paper, I am very critical.

- The english is very poor. I am not an english speaker, so I am usually quite undemanding on this aspect, but this paper really need strong rewriting. Therefore I will not give any english correction suggestions since it should be all looked up.

- Some sentences are too short, or do not seem to be at the good places, and the way the paper is designed make it quite hard to follow at some part. Illustrations will be given in the following step by step commenting of the paper.

For those main reason I recommend major revisions, and I count on the editor to verify that the scientific content worth the work to be publish.

Detailed comments:

- Abstract lines 20-21: I don't understand the sentence very well, or it feels like it is not at the good place... Shouldn't you describe your results before mentioning perspectives?? lines 33-34: In what is it pertinent? Maybe give a few clues. Line 35: "narrow range" of which parameter?

- Introduction Line 41: about reference to Löwe et al. 2011. I think that many different authors characterized snow before Löwe et al. 2011, so please find appropriate original references. (or put no reference if the info is very general) Lines 56-57: the sentence is not clear. Maybe use another term that grains that can refer to snow when dealing with a structure of the water content. + "cause" -> because of ? Line 61: The transition point in terms of water content? Temperature? etc? Please be more precise Line 76:

what does the "right" site mean? Not clear for people who don't know the curve... Line 123-125: Why is this sentence in the middle of the three regime listing?? Furthermore, only very few is said about this micro CT analysis in the paper... so either there is more about it in the paper, or you should maybe even remove it.

- Experimental setup Lines 146-147: this sentence is really unclear (english?) Line 153: what is the "star point"? Line 154: do you mean "preventing circular current to occur in the circuit"? Part 2.1: how do you preserve the liquid content during the 2.7 h of scan at -5°C? what is the sample temperature during the scan and how do you verify that there is no evolution of the sample during those 2.7 h? How did you transfer the sample from the dielectrical measurement devise to the microCT holder? There is a lot missing in this part for the reader to understand the specific methodology followed.

- Method First: please specify "Method" for what? You already presented method for microCT just before for instance.

- Results General: Make it clearer how you highlight the transition in the various signals between no percolation - percolation. It will help to access to the main result of a percolation transition for a typical water volume fraction of 5 to 12. This part 3 should be made clearer about the way the different parameters measured lead to the main result (that appear to me to be the value of the water content at the transition between no percolation and percolation). Line 216- 222: This part does not present results but the sample preparation method, please move it to the "method" part. Line 223-224: I don't understand this sentence. Isn't it what you want to obtain and to measure, in order to follow changes in those properties by measuring electrical properties?? Do you mean "Electrical properties vary with temperature, water content and present different evolution depending on the initial snow density"? The relation between electrical properties, temperature and water content are nevertheless expected by the relations provided in part 3... Line 229-230: what do you mean? The temperature profile shows the temperature increase... but not the process. Maybe "resulting from the heating process" is what you mean? Line 242: "affected", not clear, do you mean "impacted"? Line

272-273: I think that it is not "the temporal evolution" that increases but the liquid water mass and volume fraction, please rephrase. Paragraph lines 269-279: This part should be given before providing the result about the range of volume fraction corresponding to the beginning of percolation...

- Discussion Line 292: Please remind what those results are. A figure and a table is not a summary of major results, it is just the data used to access to the results... Lines 303-304: it should be the contrary!! The measured values (what values are we talking about?) depend on temperature and snow structure. And since you are heating, wouldn't it be more correct to say : " the heating process up to 0°C shows that the measured values evolve with temperature and snow structure (by the way, how can you verify that the snow structure is evolving?). Unless you mean that it depends on the density. But it is not the same process as the dependance on the temperature since temperature is evolving during the process, while the structure (density let's say) is a initial parameter not measured during the heating... Well, as you can see, I don't understand what you mean and it seems to me that you refer to a parameter (structure) that you are not evaluating. Lines 303-314: This sounds more like your interpretation rather than a demonstrated results. How do you assess these hypotheses related to the structure changes that you don't observe? Are there any previous work that could strengthen your hypotheses? And maybe say that it is an hypothetical explanation... Line 316: which "important material properties" are you talking about? Line 339: "our results", which results? please be precise. Line 346: Please discuss micro CT observations in the "results" part, in particular to be able to provide some clues on the condition that enable to be confident on what is observed (how to maintain the liquid layer during 2.7 h at -5°C, and also to show if the structure evolves (or not) during the scan, the limits due to the low resolution, etc... ). A specific part about the micro CT measurements is strongly missing.

- Summary and conclusions This part is not really a conclusion, but more a repetition of the discussion... Please be clearer about the main "take home message" that you want

to highlight in your conclusion. Line 375-377: I do not agree, I don't see where you measured the morphological properties of snow during your experiment. Apparently you deduced it from your electrical parameter measurements, but it is not clear. Please be clearer about it. The morphological observation of 1 sample after the test, that is on top of that too weakly presented, can not be used to convincingly talk about morphological evolution. Line 378-379: why don't you give a curve of the percolation starting point as a function of initial density, that would indeed make your main result turn out much clearer!

---

## Author Comment (AC2) · 12 Aug 2020

**RESPONSE TO MAURINE MONTAGNAT**
**TO MANUSCRIPT tc-2020-56-RC2**

***Title:*** Liquid-water content and dielectric properties of wet snow using microwave heating

**Authors:** Pirmin Philipp Ebner, Aaron Coulin, Joël Borner, Fabian Wolfsperger, Michael Hohl, and Martin Schneebeli

We thank Maurine Montagnat for her constructive comments and suggestions. All line numbers correspond to the discussion paper and all added texts to the discussion paper are marked red.

**MAURINE MONTAGNAT**

This paper provides interesting results about the progressive wetting of snow during heating at 0°C. To do so, dielectric properties are measured during microwave heating, and these properties enable to follow the evolution of heating during 3 stages, dry snow heating, wet snow heating, and the early stage of percolation. Thanks to the experimental configuration, these measurements enable to provide a precise analysis of the percolation initiation stage, and the corresponding water content for various initial snow densities. This seem to be the main result of the paper. An observation of the wet snow sample is done by micro computed X-ray diffraction tomography, but the results are only very briefly analysed.

Since my knowledge regarding wet snow, percolation, and dielectric measurements of water content is very weak, I am not able to estimate the relevance of the results provided, as regard to the concerned community, and to previous work. I hope that the editor or the other reviewer will be able to do so in order to complement my review work.

Regarding the way the results are presented and the organisation of the paper, I am very critical.

- The english is very poor. I am not an english speaker, so I am usually quite undemanding on this aspect, but this paper really need strong rewriting. Therefore I will not give any english correction suggestions since it should be all looked up.

- Some sentences are too short, or do not seem to be at the good places, and the way the paper is designed make it quite hard to follow at some part. Illustrations will be given in the following step by step commenting of the paper.

For those main reason I recommend major revisions, and I count on the editor to verify that the scientific content worth the work to be publish.

**Comment #1.1:** Lines 20-21: I don't understand the sentence very well, or it feels like it is not at the good place... Shouldn't you describe your results before mentioning perspectives??

> *[ANSWER] We removed this sentence and updated the abstract*
>
> Snow exists in a wide range of temperatures and around its melting point snow becomes a three-phase material. A better understanding of wet snow and the starting point of water percolation in seasonal snowpacks is essential for snow pack stability, snow melt run-off and remote sensing. In order to induce and measure precisely the liquid water content inside a snow sample, an experimental setup was developed. Using microwave heating at 18 kHz allowed to dissipate heat homogeneously through the entire volume of snow. The heating process mainly depended on initial snow density, snow temperature and liquid water content. Electrical monitoring was proved as a capable method to retrieve water content and water distribution in a snowpack. The starting point of water percolation was between 3-13 water volume fraction for snow density between 420 kg m$^{-3}$ and 620 kg m$^{-3}$. These findings are pertinent to the interpretation of the snow melt run-off of spring snow as the restrained amount of water in the snowpack can be extracted and the expected amount of water run-off into the rivers can be calculated. This experimental setup allows to find the starting point of water percolation in coarse-grained snow and to extract the corresponding dielectric properties which are important for remote sensing.

**Comment #1.2:** Lines 33-34: In what is it pertinent? Maybe give a few clues. Line 35: "narrow range" of which parameter?

> *[ANSWER] We changed the sentence to:*

L33: "These findings are pertinent to the interpretation of the snow melt run-off of spring snow as the restrained amount of water in the snowpack can be extracted and the expected amount of water run-off into the rivers can be calculated. This experimental setup allows to find the starting point of water percolation in coarse-grained snow and to extract the corresponding dielectric properties which are important for remote sensing."

**Comment #2.1:** Line 41: about reference to Löwe et al. 2011. I think that many different authors characterized snow before Löwe et al. 2011, so please find appropriate original references. (or put no reference if the info is very general)

**[ANSWER]** *We removed the reference as the info is very general.:*

**Comment #2.2:** Lines 56-57: the sentence is not clear. Maybe use another term that grains that can refer to snow when dealing with a structure of the water content. + "cause" -> because of ?

**[ANSWER]** *We changed the sentence to:*

L56: "When liquid water occurs for the first time in the seasonal snowpack, much water can be hold in the pores because of the high capillary forces of the finer microstructure."

**Comment #2.3:** Line 61: The transition point in terms of water content? Temperature? etc? Please be more precise

**[ANSWER]** *We changed the sentence to:*

L61: "The time when the liquid water …"

**Comment #2.4:** Line 76: what does the "right" site mean? Not clear for people who don't know the curve...

**[ANSWER]** *We changed the sentence to:*

L76: "… from the wetting curve."

**Comment #2.5:** Line 123-125: Why is this sentence in the middle of the three regime listing?? Furthermore, only very few is said about this micro CT analysis in the paper... so either there is more about it in the paper, or you should maybe even remove it.

*[ANSWER] We removed this sentence. Further, we added the following sentence to the end of the paragraph to provide more information about the microCT analysis:*

L126: "Additionally, we performed micro computed tomography (micro-CT) measurement to show the possibility to quantify the water content in three-dimensional space without destroying the snow structure."

**Comment #3.1:** Lines 146-147: this sentence is really unclear (english?)

*[ANSWER] We changed the sentence to:*

L146: "The generated sinusoidal waveform of voltage $U$(t) is transmitted to a differential probe before it is applied to the copper-plates."

**Comment #3.2:** Line 153: what is the "star point"?

*[ANSWER] A star point (electrical engineering) is a common junction connected to the ends of windings of a polyphase electrical device. As it is a common name in electrical engineering we don't want to provide more information.*

**Comment #3.3:** Line 154: do you mean "preventing circular current to occur in the circuit"?

*[ANSWER] We changed the sentence to:*

L154: "… preventing circular current to occur in the circuit.

**Comment #3.4:** Part 2.1: how do you preserve the liquid content during the 2.7 h of scan at -5°C? what is the sample temperature during the scan and how do you verify that there is no evolution of the sample during those 2.7 h? How did you transfer the sample from the dielectrical measurement devise to the microCT holder? There is a lot missing in this part for the reader to understand the specific methodology followed.

*[ANSWER] As mention in the text, we shock frozen the wet snow sample at -30 °C to preserve the liquid water on the snow structure, before scanning it a second time. Additionally, as the snow sample has a density of around 680 kg m-3 we assume that the evolution of the sample is low and allows us to easily transfer it from the experimental setup to the micro-CT. Also, we modified the*

*sample holder. We changed part 2.1 as follow and moved it to the method section:*

Part 2.1: "A micro-computer tomograph (CT; Scanco Medical $\mu$-CT80) at a cold laboratory temperature of -5 °C was used to visually quantify the water content in three-dimensional space without destroying the snow structure. A special modified sample holder was used (see Fig. 2) for the micro-CT scans to easily handling the transfer process between experimental setup and micro-CT. Each scan took around 2.7 h. Absorption by water and ice are almost identical (Lieb-Lappen et al., 2017), and are hardly to separate in the segmentation process. Therefore, the water creation on the snow surface was extracted by superposition of two micro-CT scans. One scan was taken before the heating process and the second one afterwards. Before the second micro-CT scan, the wet snow sample was shock frozen at -30 °C to preserve the liquid water on the snow structure. The scanned image had a volume of 200 x 200 x 20 voxels (3.6 mm x 3.6 mm x 0.36 mm) with a nominal voxel resolution of 18 µm. The grey scale resolution for each voxel was 16 bit and a Gaussian filter ($\sigma$ = 1.4, support = 3) was applied to reconstruct the micro-CT images. The volume was segmented to a binary image by classifying each voxel by ice or air. The threshold for the segmentation process was chosen such as that the manually measured density did not deviate more than 12 % from the CT-density in the segmentation process (Riche and Schneebeli, 2013). This allowed us to easily visualize and to extract the water creation on the surface of the ice matrix with an uncertainty of $\pm$2 % of liquid water mass fraction. This uncertainty is primarily caused by the segmentation process and the micro-CT resolution."

*Additionally, we add the following sentence to the discussion:*

L361: "Due to the high snow density the evolution of the snow sample can be neglected. These are first results to show the possibility to visualize the liquid water in the snow structure. However, more experiments are necessary which is not in the scope of this paper. Open questions might be answered with future experiments like: (1) using different snow densities; (2) analyse the impact of the specific surface area; (3) increase the micro-CT resolution; (4) scan the snow sample at different heating time steps; (5) multiple heating-cooling cycles; (6) use of ice beads to reduce the structural complexity.

"

**Comment #4.1:** Method First: please specify "Method" for what? You already presented method for microCT just before for instance.

*[ANSWER] We modified the method section and moved the microCT part to the method section.*

**Comment #5.1:** Results General: Make it clearer how you highlight the transition in the various signals between no percolation - percolation. It will help to access to the main result of a percolation transition for a typical water volume fraction of 5 to 12. This part 3 should be made clearer about the way the different parameters measured lead to the main result (that appear to me to be the value of the water content at the transition between no percolation and percolation).

*[ANSWER] We changed the result part and moved the highlight of the transition in the various signals between no percolation and percolation to the discussion part. In discussion part we also made clearer about the way the different parameters measured lead to the main result (starting point of water percolation).*

**Comment #5.2:** Line 216- 222: This part does not present results but the sample preparation method, please move it to the "method" part.

*[ANSWER] We moved it to the method section.*

**Comment #5.3:** Line 223-224: I don't understand this sentence. Isn't it what you want to obtain and to measure, in order to follow changes in those properties by measuring electrical properties?? Do you mean "Electrical properties vary with temperature, water content and present different evolution depending on the initial snow density"? The relation between electrical properties, temperature and water content are nevertheless expected by the relations provided in part 3...

*[ANSWER] We changed the sentence to:*

L223: "The measured electrical properties vary with temperature, water content and present different evolution depending on the initial snow density."

**Comment #5.4:** Line 229-230: what do you mean? The temperature profile shows the temperature increase... but not the process. Maybe "resulting from the heating process" is what you mean?

    *[ANSWER] We changed the sentence to:*

    L229: "The temperature profile increases from -1 °C up to 0 °C resulting from the heating process."

**Comment #5.5:** Line 242: "affected", not clear, do you mean "impacted"?

    *[ANSWER] We changed the sentence to:*

    L242: "… content strongly impacted the impedance and the …"

**Comment #5.6:** Line 272-273: I think that it is not "the temporal evolution" that increases but the liquid water mass and volume fraction, please rephrase.

    *[ANSWER] We changed the sentence to:*

    L272: "The liquid water mass and volume fraction based on the …"

**Comment #5.7:** Paragraph lines 269-279: This part should be given before providing the result about the range of volume fraction corresponding to the beginning of percolation...

    *[ANSWER] We adapted this part.*

    Line 269: " The heating efficiency was affected by heat losses at the wall and reflection of the microwave at the ice-air interface. Fig. 5 shows the heating efficiency and the complex dielectric constant of dry snow at $T = 0$ °C for various snow densities. The error bars indicate the measured uncertainty of the experimental setup. Ice had the lowest heating efficiency with the highest extracted permittivity value of $\varepsilon_i'' = 30.65$, similar to literature values of $\varepsilon_i'' = 30.93$ at 18 kHz [Fujita et al., 2000]. Reflection of microwave power at the ice-air interface, which was caused by the relative permittivity mismatch between air and ice led to limited heating efficiency. Further, our heating frequency of 18 kHz was closer to the optimal snow heating frequency of low snow densities (Bader and Kuroiwa, 1962; Polder and Van Santen, 1946; Evans, 1965) leading to better heating efficiency for lower snow densities."

**Comment #6.1:** Discussion Line 292: Please remind what those results are. A figure and a table is not a summary of major results, it is just the data used to access to the results...

> **[ANSWER]** *We removed this sentence.*

**Comment #6.2:** Lines 303-304: it should be the contrary!! The measured values (what values are we talking about?) depend on temperature and snow structure. And since you are heating, wouldn't it be more correct to say: " the heating process up to 0°C shows that the measured values evolve with temperature and snow structure (by the way, how can you verify that the snow structure is evolving?). Unless you mean that it depends on the density. But it is not the same process as the dependence on the temperature since temperature is evolving during the process, while the structure (density let's say) is a initial parameter not measured during the heating... Well, as you can see, I don't understand what you mean and it seems to me that you refer to a parameter (structure) that you are not evaluating.

> **[ANSWER]** *We changed the sentence to:*
>
> L303: "… indicated a temperature and initial snow density dependency on the measured electrical values"

**Comment #6.3:** Lines 303-314: This sounds more like your interpretation rather than a demonstrated result. How do you assess these hypotheses related to the structure changes that you don't observe? Are there any previous work that could strengthen your hypotheses? And maybe say that it is an hypothetical explanation...

> **[ANSWER]** *We changed this part:*
>
> L303-314: "The heating process up to 0 °C indicated a temperature and initial snow density dependency on the measured electrical values. As snow temperature reached the melting point, the vibration and the mobility of protons in the ice are enhanced and influenced the electrical conductivity leading to a decrease of the impedance. Further, at higher snow density the snow had more intergranular bonds increasing the permittivity (Evans, 1965). This allowed a higher rate of flow of electric charge leading to a higher electric current. Additionally, the electrical potential between the two copper-plates was less affected by the pore volume leading to a more stable voltage and smaller phase

shift between voltage and current. As a result, the electrical conductivity increased with higher snow density resulting in a lower impedance and a higher electrical energy transfer."

**Comment #6.4:** Line 316: which "important material properties" are you talking about?

*[ANSWER] We changed the sentence to:*

L316: "… changes of the electrical snow properties"

**Comment #6.5:** Line 339: "our results", which results? please be precise.

*[ANSWER] We changed the sentence to:*

L339: "… our estimated water mass volume at the starting point of water percolation were lower than found …"

**Comment #6.6:** Line 346: Please discuss micro CT observations in the "results" part, in particular to be able to provide some clues on the condition that enable to be confident on what is observed (how to maintain the liquid layer during 2.7 h at -5 °C, and also to show if the structure evolves (or not) during the scan, the limits due to the low resolution, etc... ). A specific part about the micro CT measurements is strongly missing.

*[ANSWER] We added more information to improve the micro-CT discussion:*

L348: "Due to the high snow density the evolution of the snow sample can be neglected. Further, the shock freezing before the second scan allowed us to preserve the liquid water in the snow sample."

L361: "These are first results to show the possibility to visualize the liquid water in the snow structure. However, more experiments are necessary which is not in the scope of this paper. Open questions might be answered with future experiments like: (1) using different snow densities; (2) analyse the impact of the specific surface area; (3) increase the micro-CT resolution; (4) scan the snow sample at different heating time steps; (5) multiple heating-cooling cycles; (6) use of ice beads to reduce the structural complexity."

*We added more information in the method section:*

A micro-computer tomograph (CT; Scanco Medical $\mu$-CT80) at a cold laboratory temperature of -5 °C was used to visually quantify the water content in threedimensional space without destroying the snow structure. A special modified sample holder was used (see Fig. 2) for the micro-CT scans to easily handling the transfer process between experimental setup and micro-CT. Each scan took around 2.7 h. Absorption by water and ice are almost identical (Lieb-Lappen et al., 2017), and are hardly to separate in the segmentation process. Therefore, the water creation on the snow surface was extracted by superposition of two micro-CT scans. One scan was taken before the heating process and the second one afterwards. Before the second micro-CT scan, the wet snow sample was shock frozen at -30 °C to preserve the liquid water on the snow structure. The scanned image had a volume of 200 x 200 x 20 voxels (3.6 mm x 3.6 mm x 0.36 mm) with a nominal voxel resolution of 18 µm. The grey scale resolution for each voxel was 16 bit and a Gaussian filter ($\sigma$ = 1.4, support = 3) was applied to reconstruct the micro-CT images. The volume was segmented to a binary image by classifying each voxel by ice or air. The threshold for the segmentation process was chosen such as that the manually measured density did not deviate more than 12 % from the CT-density in the segmentation process (Riche and Schneebeli, 2013). This allowed us to easily visualize and to extract the water creation on the surface of the ice matrix with an uncertainty of $\pm 2$ % of liquid water mass fraction. This uncertainty is primarily caused by the segmentation process and the micro-CT resolution."

**Comment #7.1:** Summary and conclusions This part is not really a conclusion, but more a repetition of the discussion... Please be clearer about the main "take home message" that you want to highlight in your conclusion.

*[ANSWER] We changed the conclusion part to highlight the main conclusion.*

L370: "We designed, fabricated, and tested an experimental setup for in-situ time-lapse non-destructive investigation of water percolation in snow using the electrical properties of snow. Frequency heating close to the relaxation frequency of ice was applied to slowly increase the water content uniformly in the snow sample until liquid water started to percolate. By measuring the temperature and monitoring the electrical properties, the water content in the snow sample at each timestep was deduced. This new instrument allows to

elucidating the starting point of water percolation based on measured electrical properties and initial snow density.

The presence of uniformed distributed liquid water changes the dielectric properties and reduces the impedance of the two-phase material significantly until the starting point of water percolation. After this point, the created liquid water treasures up at the bottom of the sample holder leaving bigger pores in the upper part. This leads to an increase of the overall impedance of the snow sample. At this reversal point, the start of water percolation can be calculated which is between 5-12 water volume fraction for snow density between 420 kg m$^{-3}$ and 620 kg m$^{-3}$. This setup and the obtained results can be used to precisely forecast the run-off time of different snow densities and to investigate the mechanical properties, water movements, surface friction, adhesion, and liquid-water measurements, for wet snow.

Nevertheless, our results and conclusions indicate that there is a need for additional validation. Specially, it would be crucial to not only look at the density but also at the specific surface area of the snow which also affects the capillary forces and therefore the starting point of water percolation. Additionally, more detailed micro-CT measurements are needed to make stronger statements about preferential spots of water accumulations inside the snow sample."

**Comment #7.2:** Line 375-377: I do not agree, I don't see where you measured the morphological properties of snow during your experiment. Apparently you deduced it from your electrical parameter measurements, but it is not clear. Please be clearer about it. The morphological observation of 1 sample after the test, that is on top of that too weakly presented, can not be used to convincingly talk about morphological evolution.

> *[ANSWER] We changed the sentence to:*
>
> L375: "… elucidating the starting point of water percolation based on measured electrical properties and initial snow density."

**Comment #7.3:** Line 378-379: why don't you give a curve of the percolation starting point as a function of initial density, that would indeed make your main result turn out much clearer!

> *[ANSWER] We added a new figure of the percolation starting point as a function of initial density in the discussion part and added the following text:*
> L338: "Figure 11 shows the water volume fraction as a function of density."

[Figure]

**Figure 11:** Water volume fraction at the starting point of water percolation as a function of the initial snow density. The error bars illustrate the uncertainty in the measurement.

Minor revisions were made throughout the revised manuscript.

We thank Maurine Montagna for her insight, suggestions and recommendations.

The authors